



# Landfast ice growth and displacement in the Mackenzie Delta observed by 3D time-series SAR speckle offset tracking

Byung-Hun Choe[1], Sergey V. Samsonov[1], Jungkyo Jung[2]

[1]Canada Centre for Mapping and Earth Observation, Natural Resources Canada, Ottawa, ON K1S 5K2, Canada
[2]Jet Propulsion Laboratory, California Institute of Technology, Pasadena, CA 91109, USA

*Correspondence to*: Byung-Hun Choe (byung-hun.choe@canada.ca)

**Abstract.** This study investigates the growth and displacement of landfast ice along the shoreline of the Mackenzie Delta by synthetic aperture radar (SAR) speckle offset tracking (SPO). Three-dimensional (3D) offsets were reconstructed from Sentinel-1 ascending and descending SAR images acquired on the same dates during the November 2017-April 2018 and

October 2018-May 2019 annual cycles. The results showed horizontal and vertical displacements of floating landfast ice caused by ice breakups and pressure ridges, which are mainly driven by drift sea ice motions and Mackenzie Delta discharges. Cumulative vertical offsets of approximately -1 to -2 m were observed from freshwater landfast ice, which is due to longer radar penetration into the ice-water interface with increasing landfast ice thickness. Numerical ice thickness model estimates confirmed that the cumulative vertical downward offsets indicate the growth of freshwater landfast ice thickness. Time-series

analysis showed that significant growth and displacement of floating landfast ice in the Mackenzie Delta occur between November and January.

## 1 Introduction

Recent climate change reports highlight the rapidly decreasing sea ice extent and thickness with record high temperatures in the Arctic (Kwok, 2018; Lang et al., 2017; Parkinson and DiGirolamo, 2016; Simmonds, 2015). Its decreasing maximum

extent and shorter annual cycle with later freezeup and earlier breakup indicate rapid climate change in the Arctic (Parkinson, 2014). Landfast ice is a type of sea ice formed on the land or extended from the land, which can be classified into bottomfast ice, stabilized floating ice, and non-stabilized floating ice extensions (Dammann et al., 2019). The annual maximum extent of Arctic landfast ice is ~1.8M $km^2$, which is about 12% of the Northern Hemisphere sea ice extent (Yu et al., 2014). The landfast ice thickness in the Canadian Arctic Archipelago (e.g., Cambridge Bay, Eureka, Alert) has decreased at ~4 cm per decade with

changes in snow depth (Howell et al., 2016). Landfast ice plays important roles for coastal sediment and hydrological dynamics (Eicken et al., 2005; Itkin et al., 2015), marine mammal habitats (Lovvorn et al., 2018), and traffic and hunting activities of northern coastal communities (Laidler et al., 2009). It also serves as a nearshore platform for oil and gas exploration in the Arctic (Masterson, 2009). Thus, spatial and temporal monitoring of landfast ice is critical for accessing climate change impacts and natural hazards in the Arctic.



The Mackenzie Delta located in Northwest Territories, Canada, is the second largest delta of ~13000 km$^2$ in the Arctic (Fig. 1a). The terrain is underlain by arctic permafrost of ~ 100 to 500 m thickness (Burn and Kokelj, 2009). The Mackenzie River and surrounding channels discharge freshwater of ~284 km$^3$ annually (Emmerton et al., 2007). In the Mackenzie shelf along the Beaufort Sea coast, landfast ice recurrently forms from fall or early winter and melts away by early summer. Landfast ice and drift sea ice are mainly affected by winds and ocean currents (e.g., the Beaufort Gyre specifically in this region), but they

are also impacted by heat transports from ocean mixing and river discharges (Nghiem et al., 2014; Zhang et al., 2013).

Optical sensors were used to map sea ice type and extent, but it is very limited to acquire optical imagery not obscured by cloud and atmosphere in the Arctic with the polar night of ~6 months during the winter. SAR, on the other hand, has a great advantage of all-weather and day and night imaging capability, so it has been extensively used to monitor sea and landfast ice

(Onstott, 2011). SAR backscattering can characterize ice types (e.g. multi-year vs. first-year ice), and roughness (Lyden et al., 1984; Nghiem and Leshkevich, 2007; Wakabayashi et al., 2004). SAR penetration depth is very sensitive to the salinity of ice (Hallikainen and Winebrenner, 2011). In case of landfast ice affected by significant freshwater inflow, such as the Mackenzie Delta and the Lena Delta, it has very low salinity close to zero (Macdonald et al. 1995; Eicken et al., 2005). In the Mackenzie shelf near the mouth of the Mackenzie River, freshwater landfast ice forms by incorporating river discharges spreading along

the coast (Macdonald et al., 1995). This allows C-band radar with a penetration depth of several meters for dry snow and freshwater ice to reach into the ice-water interface (Rignot et al. 2001; Yue et al., 2013), unlike saline sea ice with a very short penetration depth of tens of centimeters (Dammann et al., 2018; Hallikainen and Winebrenner, 2011). Thus, the lowering ice-water interface with the growth of landfast ice thickness can be translated into vertical changes in SAR interferometry (InSAR) and speckle offset tracking (SPO) analyses. Well-established InSAR techniques have been used to study glacier and sea ice

motion and dynamics in the Arctic and Antarctica (Goldstein et al., 1993; Rignot et al., 1995, 2011; Vincent et al., 2004). Yue et al. (2013) delineated bottomfast ice from floating landfast ice in the Mackenzie Delta by combining InSAR coherence with polarimetric SAR classification. Recently, Dammann et al. (2019) classified landfast ice in the pan-Arctic including the Makenzie Delta depending on the InSAR fringe patterns observed at the most stable ice growing end stage (Fig. 1b). While no fringes are observed from bottomfast ice, distinct fringes start to appear from stabilized floating ice and much denser fringes

are observed from non-stabilized floating ice extensions (Dammann et al., 2019). However, landfast ice is comprised of different plates and cracks. This discontinuous nature of landfast ice can limit the estimation of displacements by unwrapping the interferometric phase modulated in $2\pi$.

In this work, we investigate the vertical and horizontal changes of landfast ice in the Mackenzie Delta using Sentinel-1 SAR

data. We present InSAR observations relating to landfast ice changes and its limitations inherent to phase unwrapping of landfast ice and very low coherence during ice growth. We propose a novel 3D SAR SPO technique for monitoring the growth and displacement of landfast ice of sub-meter precision without the uncertainty of phase unwrapping, and compare the results



with climate and environmental factors contributing to landfast ice changes. We demonstrate that freshwater landfast ice thickness can be directly obtained from the 3D SPO measurements.

## 2. Sentinel-1 SAR and supporting data

A total of 18 ascending (path: 108, frame: 226) and 18 descending (path: 116, frame: 360) Sentinel-1 Terrain Observation with Progressive Scan (TOPS) SAR images of VV polarization were collected for two annual cycles of November 2017–April 2018 and October 2018–May 2019 (Table 1, Fig. 2). Sentinel-1 TOPS SAR single look complex (SLC) data were acquired at a spatial resolution of ~2.33 m by ~13.89 m in line-of-sight (i.e. slant-range) and azimuth at incidence angles of ~30˚ - 45˚. The

Sentinel-1 TOPS SAR interferometric wide (IW) mode with a ~250 km swath can acquire an overlapping spatial coverage from ascending and descending flights less than 1-day at high latitudes (Torres et al., 2012). The Polar Pathfinder daily 25 km EASE-Grid sea ice motion data (available at https://nsidc.org/data/nsidc-0116) produced from multiple sensor observations and buoy and wind measurements provided by the National Snow and Ice Data Center (NSDIC) were compared with the SPO measurements. Daily air temperature (from the Pelly Island station at 69° 63´ N, 135° 44´ W, available at

http://climate.weather.gc.ca), snow depth (from the Inuvik station at 68° 32´ N, 135° 44´ W, available at http://climate.weather.gc.ca), freshwater discharges and water level (from the Mackenzie River Middle channel (10MC008) at 68° 17´ N, 134° 25´ W, available at http://wateroffice.ec.gc.ca), and hourly tide records (from the Tuktoyaktuk station at 69° 43´ N, 132° 99´ W, available at https://tides.gc.ca) provided by the Environment and Climate Change Canada (ECCC) and Fisheries and Oceans Canada (DFO) were analyzed.

## 3. Methods

### 3.1 InSAR

InSAR measures the phase difference in line-of-sight between two SAR acquisitions. The phase difference at a cm-scale wavelength is modulated in $2\pi$, which can relate to displacement or dynamics of target surfaces. The InSAR processing was performed with the GAMMA software (Werner et al., 2000). Based on the perpendicular baselines, a master image was

selected for each annual cycle and the other slave images were precisely co-registered with the master image (Fig. 2). The interferograms were generated by multilooking with a window of 12 by 4 pixels in range and azimuth (i.e., corresponding to ~28 m by 55 m in range and azimuth), and the topographic phase for land was removed with the 30-m resolution ASTER global digital elevation model (GDEM). The multilooked interferograms were filtered by the Goldstein adaptive filter (Goldstein and Werner, 1998) and unwrapped by the minimum cost flow (MCF) algorithm (Costantini, 1998).




### 3.2 3D SPO

SAR SPO estimates the offsets of pixels (or subpixels by oversampling) in line-of-sight and azimuth directions from SAR images (Strozzi et al., 2002). The offsets are estimated by computing the cross-correlation of small image patches, and the signal-to-noise ratio (SNR) is calculated by the peak value relative to the average of the cross-correlation function for noise

removal (Strozzi et al., 2002). The line-of-sight and azimuth offsets, calculated from each of ascending and descending pairs ($LOS_i$ and $AZI_i$ ; $i=asc, dsc,$ four product sets in total), are used to reconstruct the 3D offsets ($D_j$; $j=N, E, U$) for each acquisition epoch by inverting the following Eq. (1), written in a matrix form (Fialko et al., 2001):

$$
\begin{pmatrix} LOS_{asc} \\ LOS_{dsc} \\ AZI_{asc} \\ AZI_{dsc} \end{pmatrix} = \begin{pmatrix} \sin\varphi_{asc}\sin\theta_{asc} & -\cos\varphi_{asc}\sin\theta_{asc} & \cos\theta_{asc} \\ \sin\varphi_{dsc}\sin\theta_{dsc} & -\cos\varphi_{dsc}\sin\theta_{dsc} & \cos\theta_{dsc} \\ \cos\varphi_{asc} & \sin\varphi_{asc} & 0 \\ \cos\varphi_{dsc} & \sin\varphi_{dsc} & 0 \end{pmatrix} \begin{pmatrix} D_N \\ D_E \\ D_U \end{pmatrix},
\tag{1}
$$

where $\varphi_i$ and $\theta_i$ ($i=asc, dsc$) are the azimuth and incidence angles of ascending and descending pairs, respectively. Here, for the purpose of modeling, we assume that each pair of ascending and descending data were acquired at the same time, although there is a difference of several hours. The offsets were also calculated with the GAMMA software (Werner et al., 2000) at 64 by 16 pixels spacing in range and azimuth (i.e., ~149 m by 222 m in range and azimuth) using a search window of 256 by 64 pixels in range and azimuth. A threshold of SNR=5 and the median filter with a window size of 11 pixels in range and azimuth

were applied to remove noise. The standard deviations of the cumulative offset estimates for north-south, east-west, and up-down components were ~0.4 m, ~0.3 m, and ~0.2 m, respectively. Time series analysis was performed for each of 2017-2018 and 2018-2019 annual cycles by applying the Small Baseline Subset (SBAS) technique using MSBAS software (Samsonov and d'Oreye, 2017).

## 4 Results and discussion

### 4.1 InSAR observation and limitation

Similarly to Dammann et al. (2019), we observed distinct InSAR fringe patterns relating to landfast ice displacements with an ascending pair of 20170319-20170331, which were confirmed by Landsat 8 true color composites showing landfast ice breakups occurred during the overlapping time period (Figs. 3a and 3b). Compared to the bottomfast ice, where radar signals penetrating into grounded ice are mostly absorbed into ground (i.e., very weak backscattering), much stronger backscattering

responses were observed from the stabilized floating landfast ice adjacent to the coastline (Figs. 3c and 3d). This suggests that part of the stabilized floating landfast ice in Fig. 1b is freshwater ice that allows C-band SAR to penetrate and the strong signals are backscattered from the ice-water interface (Hirose et al., 2008; Jeffries et al., 1996; Yue et al., 2013). Beyond the outer edge of the freshwater landfast ice, distinctly weak backscattering responses were observed. This indicates saline floating



landfast ice where radar signals are backscattered only from smooth saline ice surfaces with little penetration to ice resulting
in relatively weak backscattering responses. Macdonald et al. (1995) confirmed the different freshwater proportion and salinity
within the Mackenzie Delta landfast ice zone by analyzing ice core samples. While InSAR fringes were observed from the
outer saline part in the stabilized floating landfast ice and much denser fringes appeared from non-stabilized floating ice, the
freshwater floating landfast ice was masked out with very low coherence (Fig. 3e). In addition, InSAR analysis showed a
limitation in phase unwrapping to estimate the quantitative displacement from the 2π-modulated interferometric phase.
Different plates and cracks within landfast ice and abrupt transitions in the boundaries between land and ice showed significant
discontinuities in the unwrapped phase resulting in inaccurate displacement estimation (Fig. 3f). For 2017-2018 and 2018-
2019 cycle datasets, the coherence of floating landfast ice signals during the ice growing stage is mostly lost, which does not
allow generating SAR interferograms (Fig. 4). Also, orbital phase ramps were frequently observed from this region in the
Arctic, and they were hard to remove (Figs. 4a and 4b). Thus, it is very challenging to detect relatively coherent changes of
landfast ice by InSAR analysis. Compared to the InSAR coherence calculated with surrounding pixels, the SNR calculated
with a larger search window was significantly maintained over the floating landfast ice (Figs. 4g-4i), which shows a great
potential for monitoring landfast ice changes even during its growth.

## 4.2 3D SPO and time series analysis

Fig. 5 shows the 3D SPO results for landfast ice during the 2017-2018 cycle compared to the average drift sea ice daily motions
during the same time period. The stabilized floating landfast ice characterized by distinct interferometric fringe patterns (saline
ice) and strong backscattering (freshwater ice) in Fig. 3 showed significant horizontal and vertical offsets. The horizontal
offsets are caused by the lateral displacement of the floating landfast ice. In January 2018, the floating landfast ice showed
horizontal displacements towards northwest, which are consistent with the direction of Mackenzie Delta discharges towards
Beaufort Sea and the drift sea ice motions heading to west along the coastline (Fig. 5a). Vertical downward offsets of <0.5 m
were observed along the Beaufort Sea around the Mackenzie River mouth. On the other hand, vertical upward offsets were
observed along the seaward edges. These edges correspond to where open water leads occur with recursive ice freezing and
breakup, and SPO was not applicable beyond the edges due to faster motions (i.e., non-stabilized floating ice, masked out by
the threshold of SNR). In late January to early March 2018, sudden upward vertical offsets of >0.5 m were observed during
the 12- and 24-day intervals (Figs.5b and 5c). These are considered to be pressure ridges formed by the collision between
landfast ice and drift sea ice. The drift sea ice motions were confirmed to move towards the land from the north during the
same periods. Overall, the floating landfast ice changes for the 2017-2018 cycle are characterized by the horizontal offsets
heading to northwest and the vertical downward offsets expanding towards the Beaufort Sea coasts out of the Mackenzie River
mouth (Fig. 5d).





The 3D time-series analysis confirmed the cumulative horizontal displacements up to ~8 m towards north and west and the
       cumulative vertical offsets of up to ~-2.3 m from the floating landfast ice (Fig. 6). Three points from the floating landfast ice
       showing different fringe patterns according to the distance from the shore were analyzed with a reference point at the land.
       The horizontal displacements increase approaching to the seaward edge. LFI 1 formed close to the land showed little variation
       in north-south and east-west offsets similarly to the land, but only vertical offsets of ~-1 m (Fig. 6). LFI 3 close to the seaward

edge showing the largest changes could not be estimated in the 2018-2019 cycle due to faster movements (i.e., low SNR). The
       horizontal displacements towards north are likely affected by freshwater discharges from the Mackenzie River and surrounding
       channels flowing into the Beaufort Sea. It is reported that the average amount of freshwater discharges in this region is ~9000
       to ~13000 $m^3$/s in early September and decreases to ~3500 $m^3$/s in January. Since January, the lowest amount is maintained
       until late April and rapidly increases from early May. The horizontal displacements towards west correspond to drift sea ice

motions mostly heading to west, which are driven by wind and ocean currents. The horizontal and vertical changes were
       predominant between November and January and were larger in the 2018-2019 cycle.

       The vertical downward offsets are from longer radar penetration into the ice-water interface as landfast ice grows. The snow
       depth has steadily increased reaching ~0.6 m until late April. The snow layer in cold and dry conditions over the winter are

transparent to C-band SAR (see Figs. 3c and 3d), so this cannot contribute to the downward offsets. The local freshwater level
       in the Mackenzie Delta has decreased until late November and then has increased from early December, but the variation was
       small and the local freshwater level changes have little affect on the Beaufort sea level. Also, tide records showed very small
       fluctuations between ±0.3 m during the time periods, so it also cannot affect the steady downward offsets.

### 4.3 Comparison with numerical ice thickness modeling

The amount of the cumulative vertical offsets is comparable to the ice thickness measured at ~1 to 2 m with a ground
       penetrating radar in the region (Solomon et al., 2005). The reversed cumulative vertical offsets correspond to the growth
       patterns of landfast ice thickness observed in the Canadian Arctic Archipelago (Howell et al., 2016). Thus, we converted the
       cumulative downward vertical offsets into the relative growth of ice thickness (i.e. reversing a negative vertical offset to a
       positive growth of ice thickness), and compared them to ice thickness model estimates (Fig. 7). We applied a sea ice thickness

model based on accumulated freezing degree days (AFDD) developed by Lebedev (1938), Eq. (2):

$$h_i = \beta * \left[ \int (T_f - T_a) dt \right]^{\gamma}, \tag{2}$$

       where $h_i$ is the ice thickness, and $T_f$ and $T_a$ are the freezing point temperature and the air temperature, respectively (King et al.,
       2017). $\beta$ and $\gamma$ are empirical coefficients varying depending on the target environments such as average snow depth, and $\beta$
       ranging from 1.7-2.4 and $\gamma$ ranging from 0.5-0.6 have been tested at different sea and lake ice sites (Murfitt et al., 2018). We

calculated AFDD with the daily mean air temperature and $T_f$=0 °C. We used $\beta$=0.94 and $\gamma$=0.6 optimized for estimating



freshwater lake ice thickness (Model 1, Murfitt et al. 2018). The growth of LFI 1 close to the land matched well with Model 1 estimates. LFI 2 and LFI 3 close to the seaward edge showed faster growth, particularly in November to January, and reached up to ~2.5 m matching with optimized $\beta=2.25$ and $\gamma=0.6$ (Model 2).

## 5 Conclusions

We developed a methodology for monitoring landfast ice displacement and, in some cases, ice thickness. Ice thickness can be measured only for freshwater ice that allows SAR to penetrate and where backscattering occurs at the ice-water interface. While the InSAR observations showed very low coherences over landfast ice between SAR acquisitions and discontinuity problems in converting the interferometric phase into displacement, the 3D SAR SPO showed a greater potential. The question remains why high SNR values are observed in the areas with low coherence. Possibile explanations are that a significantly

larger window is applied to calculate for SNR than for coherence, and large deformation gradients can produce low coherence. Probably both reasons are valid. This will be further addressed in follow-up studies. Horizontal and vertical displacements caused by ice breakups and pressure ridges were observed, which correspond to drift sea ice motions driven by wind and ocean currents. The horizontal displacements are also largely affected by freshwater discharges from the Mackenzie Delta. The cumulative vertical offsets indicated the growth of freshwater landfast ice thickness except the early freezing stage and the late

melting stage of very rapid changes. The time-series analysis revealed the most significant growth and displacement of landfast ice occur between November and January. This is because the amount of freshwater discharge is much larger during the early season and decreases until January, and then a very small amount of freshwater discharge continues until late April. The proposed methodology can be used to estimate the thickness of freshwater landfast, lake, and river ice. Further study needs to be extended to the pan-Arctic landfast ice mapping with more Sentinel-1 coverages and a new coming trio of RADARSAT

Constellation Mission (RCM) SAR satellites with a 4-day revisit cycle. Higher spatial and temporal resolution SAR with a capability to acquire at the same time from ascending and descending orbits can greatly improve the precision of SPO measurement.

*Author contributions.* Byung-Hun Choe conceived the research, conducted SAR data processing and modeling, and wrote the original manuscript. Sergey Samsonov provided critical guidance on all aspects of SAR data processing and analysis. Jungkyo

Jung contributed to InSAR data processing and analysis. Both co-authors reviewed and edited the manuscript.

*Competing interests.* The authors declare no conflict of interest.

*Acknowledgements.* This work was funded by the Canadian Space Agency (CSA) through the Data Utilization and Application Plan (DUAP) program. Sentinel-1 data were provided by the European Space Agency (ESA), and drift sea ice daily motion



data were provided by the U. S. National Snow and Ice Data Center (NSDIC). Air temperature, snow depth, tide, and freshwater
discharge records were provided by the Environment and Climate Change Canada (ECCC) and Fisheries and Oceans Canada
(DFO).

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




Table 1: Sentinel-1 SAR datasets for 2017-2018 and 2018-2019[*]

| 2017-2018 dataset | $B_T$ (days) | 2018-2019 dataset | $B_T$ (days) |
|---|---|---|---|
| 20170319-20170331 | 12 | 20181028-20181203 | 36 |
| 20171126-20180113 | 48 | 20181203-20190108 | 36 |
| 20180113-20180125 | 12 | 20190108-20190120 | 12 |
| 20180125-20180206 | 12 | 20190120-20190201 | 12 |
| 20180206-20180302 | 24 | 20190201-20190225 | 24 |
| 20180302-20180314 | 12 | 20190225-20190309 | 12 |
| 20180314-20180326 | 12 | 20190309-20190321 | 12 |
| 20180326-20180407 | 12 | 20190321-20190402 | 12 |
| | | 20190402-20190414 | 12 |
| | | 20190414-20190426 | 12 |
| | | 20190426-20190508 | 12 |

* $B_T$: temporal baseline

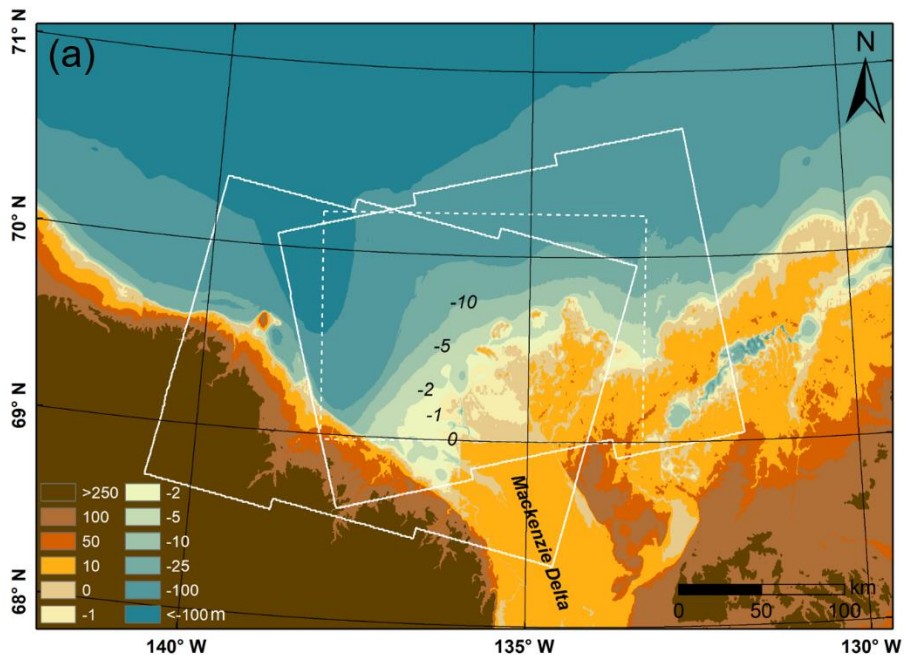

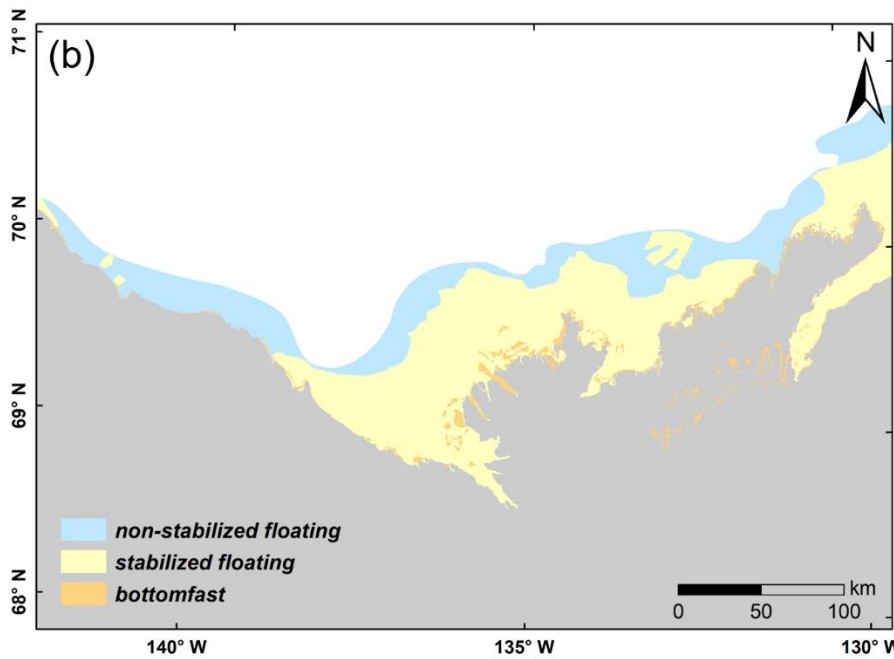


**Figure 1.** Digital elevation model of the Mackenzie Delta (a) and InSAR-derived landfast ice map (b, modified from Dammann et al. (2019), land is masked out in grey). The solid white lines are Sentinel-1 ascending and descending coverages used in this study. The dashed white rectangle represents the coverage of Sentinel-1 and Landsat-8 subsets in Figs. 3 and 4. The DEM was modified from the coastal digital elevation model global mosaic provided by National Oceanic and Atmospheric Administration (NOAA) / National Centers for
Environmental Information (NCEI). The 3D SPO in Fig. 5 was performed for the overlapped part of ascending and descending coverages.



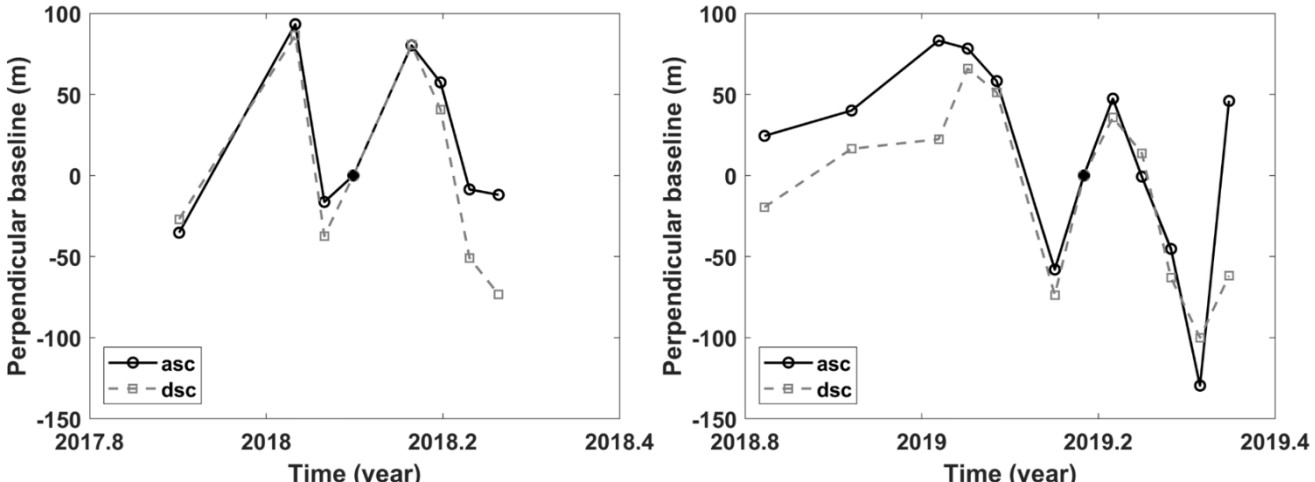

**Figure 2.** Perpendicular and temporal baseline plots of Sentinel-1 SAR datasets in Table 1 (left: 2017-2018 cycle, right: 2018-2019 cycle). The filled markers represent the master images.

**Figure 3.** Landsat 8 and Sentinel-1 observations of landfast ice in the Mackenzie Delta. (a) Landsat 8 true color composite of 20170313 (yyyymmdd). (b) Landsat 8 true color composite of 20170329 showing open water leads after a breakup. (c) SAR intensity image of 20170319. (d) SAR intensity image of 20170331. (e) SAR interferogram of 20170319-20170331 overlaid on the intensity image of 20170319. (f) Unwrapped phase showing discontinuities (black arrows) of 20170319-20170331. The SAR intensity images were linearly stretched between -25 dB and -5 dB. The orange arrows represent bottomfast ice. The green, light yellow, and light blue dashed lines with double arrows represent the ranges of stabilized floating freshwater ice, stabilized floating saline ice, and non-stabilized floating saline ice, respectively. The coastline is shown in yellow (c, d) and black (e, f).

**Figure 4.** Examples of Sentinel-1 SAR interferograms (a: 20180113-20180125, b: 20180125-20180206, c: 20180206-20180302) and their coherence (d: 20180113-20180125, e: 20180125-20180206, f: 20180206-20180302) and SNR (g: 20180113-20180125, h: 20180125-20180206, i: 20180206-20180302).



**Figure 5.** 3D SPO results. (a) 20180113-20180125. (b) 20180125-20180206. (c) 20180206-20180302. (d) Total cumulative offsets between 20171126 and 20180407. The color bars represent vertical offsets and the black arrows represent horizontal offsets reconstructed from east-west and north-south offsets. The pale blue arrows represent the averaged drift sea ice motions during the same period. Note that the sizes of the arrow scale bars vary relative to the changes for each period. The black (land) and red (landfast ice; LFI 1-3) circles represent the spots for time-series analysis in Fig. 6.





**Figure 6.** 3D time-series analysis of 2017-2018 ((a) north-south, (c) east-west, (e) up-down) and 2018-2019 ((b) north-south, (d) east-west, (f) up-down) cycles. The green, orange, yellow, blue lines represent the offsets observed from the land and landfast ice (LFI 1-3) spots marked in Fig. 5d, respectively. The vertical error bars represent the standard deviation in 5 by 5 pixels window.





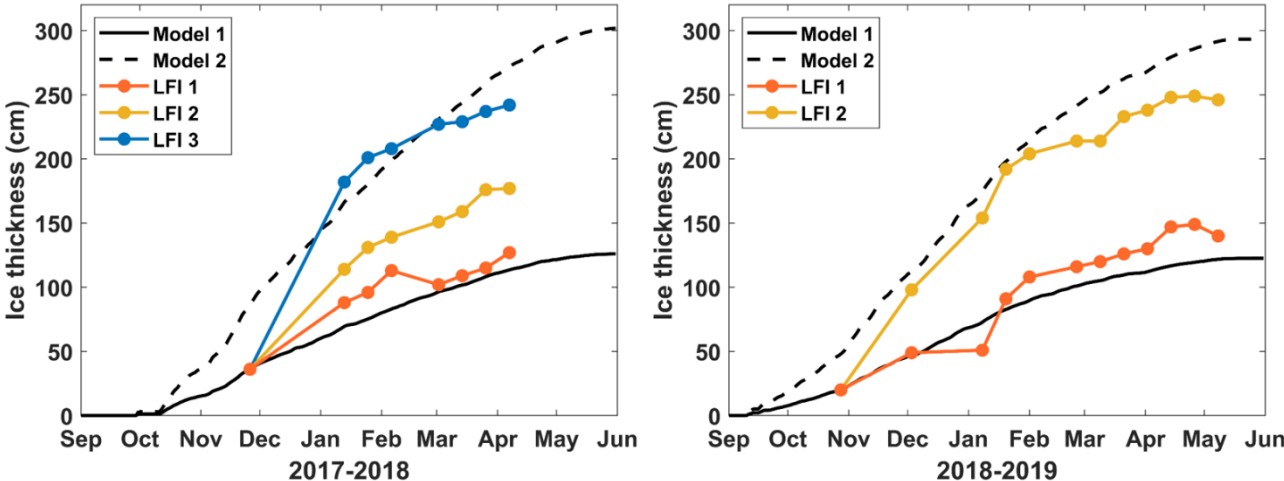

380 **Figure 7.** Comparison between ice thickness model estimates and vertical offsets observed from SAR SPO (left: 2017-2018, right: 2018-2019). The black solid and dashed lines are the estimates by Model 1 and Model 2, respectively. The orange, yellow, and blue markers represent the relative ice thickness reversed from the vertical offsets of LFI1, LFI2, and LFI3 in Figure 4, respectively. The SPO starting points are referenced to the Model 1 estimates of the first SAR acquisition dates.