# Peer review of "Landfast ice growth and displacement in the Mackenzie Delta observed by 3D time-series SAR speckle offset tracking"

_The Cryosphere, 2020_

## Referee Comment (RC1) · Dustin Whalen (Referee) · 9 Jun 2020

This is a very interesting paper presenting a new innovative method for the growth and displacement of landfast ice from SAR. The information was well presented and addressed an important issue of landfast ice growth in the Mackenzie Estuary. Field observations would increase the value of this paper but their absence in this case is quite understandable given the extreme remoteness and accessibility of the site.

Air temperature is an important component of the methodology towards the sea thickness model but the authors fail to mention what the trend was during the study period. It would be good to know how this compares with the information presented

in the discussion. Is there a threshold of growth once a maximum temperature is reached? Is that why maximum growth takes place between November to January and not after.

Please also note the supplement to this comment:
https://www.the-cryosphere-discuss.net/tc-2020-116/tc-2020-116-RC1-supplement.pdf

**Supplement:**

General Comments

This is a very interesting paper presenting a new innovative method for the growth and displacement of landfast ice from SAR. The information was well presented and addressed an important issue of landfast ice growth in the Mackenzie Estuary. Field observations would increase the value of this paper but their absence in this case is quite understandable given the extreme remoteness and accessibility of the site. As mentioned in the manuscript, I believe this information could have further practical applications to be used with the oil and gas industry and increase our knowledge of ice interactions in a changing climate. A better understanding of ice conditions is critical for future development and safety concerns. I also wonder if this method could be adopted for rivers (like the Mackenzie), as ice thickness can increase the chances of mechanical break-up leading to ice jamming and community flooding.

Specific Comments

The author mentions drift, wind and ocean currents as a possible reasoning for landfast ice displacement, but very little evidence is presented to back up this claim other than reference to ice sheet break-up shown in Figure 3. Perhaps a wind rose, or a paragraph of winds statistics for the study time period may help to highlight this point.

Air temperature is an important component of the methodology towards the sea thickness model but the author failed to mention what the trend was during the study period. It would be good to know how this compares with the information presented in the discussion. Is there a threshold of growth once a maximum temperature is reached? Is that why maximum growth takes place between November to January and not after.

Did the authors look at the growth and displacement of bottomfast ice during the study period; does it show comparable growth patterns between the years?

Technical Corrections

The figures were well done and helped to explain the results. A few edits below:

Figure 1 A – elevation model on land appears to be discontinuous with the abrupt break in the elevation at the latitude line. I am also confused on the colour, do they represent a range? (ie. 0-10, 10-50, 50-100). I think the map would look more realistic if you tighten the range of elevations at the lower levels (ie. 0-2, 2-5, 5-10, 10-50, etc) Bathymetry estimates seem believable.

Figure 1: (caption): missing bracket after b, adjust other brackets in caption to equal out.

Figure 5b: Please explain the large offset arrow coming from the Yukon coast in the Shingle Point area.

Figure 5d: The red circles are hard to see

Figure 6: Perhaps the land value can be black which is instantly distinguishable and matches the circle in Figure 5.

Line 35: Does bathymetry play a role in the distribution of landfast ice – perhaps it should be mentioned here. The shallow water depth of the Mackenzie Estuary definitely plays a role in the distribution of bottomfast ice and subsequent water discharge through these regions defined by the bottomfast ice barrier.

Line 129: Please explain what an orbital phase ramp is?

Line 158: Please add reference

160 – I think it would be good to insert some stats on prevenient wind direction from the Pelly station.

168, Yes agree, tidal fluctuations are nil but wind driven storm surge (even in the winter) have been known to cause over ice flooding. It would be good to know what the wind pattern was during the study period.

171 – Chris Stevens (University of Calgary 2006-2011) conducted a MSc and PHd looking at the this area through GPR, perhaps his subsequent publication would provide some validation.   May I also suggest the GSC online expedition database ED at Sea for further validation? https://ed.gdr.nrcan.gc.ca/index_e.php   search cruise 2007301, Solomon describes a series of ice measurements with thickness and bottomfast ice.

---

## Referee Comment (RC2) · Anonymous Referee #2 · 10 Jun 2020

**Landfast Ice Growth and Displacement in the Mackenzie Delta Observed by 3D Time-Series SAR Speckle Offset Tracking**
By B.-H. Choe and others
*Submitted to The Cryosphere*

*Review*
*June 9, 2020*

Summary
This manuscript is nearly identical to a manuscript by the same authors that I recently reviewed for Geophysical Research Letters. Unfortunately I see that my fundamental concerns regarding the feasibility of measuring ice growth with this technique have not been addressed and the authors have not considered the alternative and simpler mechanism that I proposed to explain the apparent vertical motion of the scattering surface of the ice. As a result, my review remains substantially similar to the review I provided when this manuscript was submitted to GRL.

The manuscript presents an interesting analysis of SAR speckle offset tracking applied to the measurements of horizontal ice motion and ice growth in landfast ice near the Mackenzie Delta. By combining results from imagery acquired from same-day ascending and descending orbits, the authors are able to estimate 3D motion of the surface from which radar energy is backscattered. In the case of the relatively low-salinity ice found in parts of the Mackenzie Delta, the scattering interface is assumed to be the ice-water interface and thus the observed vertical motion of the surface is interpreted as ice growth. This finding is supported by results from a 1D ice growth model which is in agreement with observed downward motion of the scattering surface. Elsewhere, positive vertical motion is assumed to indicate pressure ridging. Significant horizontal motion is also observed, which is attributed to wind, currents and river discharge.

The authors have chosen a complex region of the Arctic for their study, where atmospheric, cryospheric, marine and terrestrial processes interact. Unfortunately, the text suggests that the authors do not have a deep familiarity with the geophysics of these systems and as a result I fear they are misinterpreting their results. In particular, I am skeptical that the backscatter in the regions where the ice exhibits downward motion is coming from the bottom of the ice as the authors assume. Details are given in my major comments below, but in short, my reasons are:
  i.    The backscatter from the ice bottom is unlikely to remain coherent over periods of 12 days or more during growth
  ii.   the authors present no direct observations of the ice salinity to support the assertion that the C-band radar is penetrating to the bottom of the ice
  iii.  the authors overlook a much more likely mechanism for downward motion of the ice surface near a large delta during winter.

3D measurements of small-scale ice motion could be of considerable value for understanding dynamical processes in the Arctic coastal zone and I do not wish to discourage the authors from

continuing this line of research. However, if they must address my concerns below if they are going to continue to assert that their observations are related to thickening of the ice cover.

**Major Comments**

1. No explanation of why backscatter from ice bottom would remain coherent during growth
The speckle offset tracking technique requires that the scattering surface remains coherent between image acquisitions. However, in the case of scattering from the bottom of a growing floating ice cover, the radar is seeing an entirely new surface at each acquisition and I therefore see no reason why the speckle would remain consistent over timespans of 12 days. The authors need to provide more explanation of how the scattering characteristics of the underside of the sea ice (if indeed that is where the signal is coming from) would remain constant as new ice forms below each previously imaged surface.

2. Inadequately supported assumption that ice bottom is source of SAR backscatter
The authors assume that the SAR signal from the stabilized floating landfast ice is coming from the ice-water interface. The basis of this assumption is the low-backscatter signature of presumed bottomfast ice nearby and the presence of low-salinity ice in this region reported by MacDonald et al (1995). However, none of ice sampled by MacDonald et al was completely fresh and most contained a significant seawater fraction. Moreover, close inspection of the SAR intensity imagery (Fig 3d,e) shows linear spatial patterns typically associated with surface roughness features. It therefore seems likely that some fraction of the microwave energy returned from the ice is coming from its upper surface and volume. This has a significant bearing on the interpretation of the SPO results, but is not discussed in the manuscript.

3. Elevation increase due to ridging is unlikely to be detectable with this method
The authors attribute positive vertical ice surface motion to be the result of pressure ridging. This explanation sounds highly unlikely since any such ridging would dramatically change the surface scattering characteristics of the ice such that coherence would be lost between image acquisitions. This is similar to the problem of maintaining coherence from the ice bottom during thermodynamic growth, but a more extreme example of surface change.

4. No discussion of vertical motion due to changes in local water level
I was surprised that the authors failed to consider other sources of vertical motion, besides ice growth and ridging. A far more likely source of vertical motion is variation in local sea level due to tides, winds, currents and river discharge. The authors clearly state that the discharge from the Mackenzie River continues to decrease between January and April, but fail to consider that this might lead to a decrease in water level near the Delta. This is a far more likely explanation for the negative vertical motion at locations LFI 1-3 that does not require the radar to penetrate the ice or for speckle to remain consistent from a growing ice bottom.

5. Interpretation of river current-induced horizontal motion is unsupported
The text repeatedly makes a connection between offshore motion of the landfast ice near the Mackenzie Delta and the direction of river discharge toward the Beaufort Sea (e.g., lines 137-139; lines 155-157; and line 193). However, I find this an unlikely explanation and the only

supporting evidence the authors provide is that the discharge in winter is non-zero. If the authors wish to strengthen their claim that any offshore motion of the landfast ice caused by river outflow, they should estimate the likely basal stress on the ice and describe a likely mechanism by which this could fracture and displace the ice.

---

## Author Comment (AC1) · 5 Aug 2020

**Responses to RC1**

The author mentions drift, wind and ocean currents as a possible reasoning for landfast ice displacement, but very little evidence is presented to back up this claim other than reference to ice sheet break-up shown in Figure 3. Perhaps a wind rose, or a paragraph of winds statistics for the study time period may help to highlight this point.

Agreed. An additional figure of wind rose graphs was produced to show the wind statistics during the 2017-2018 period (Figure S1). We confirmed the horizontal displacements are largely affected also by wind, not only by ocean currents. In particular, the distinct northwest displacements are considered to be due to the strong SE wind from the Mackenzie Delta. Also, the horizontal displacements during 20180125-20180206 (Figure 5b) and 20180206-20180302 (Figure 5c) were affected by the strong W and WNW winds. This new figure will be added after Figure 5 and explained in the revised manuscript.

[Figure]

*.Figure S1. Wind statistics (Pelly Island station). (a) 20180113-20180125. (b) 20180125-20180206. (c) 20180206-20180302. (d) 20171126-20180407. North is 0°*

Air temperature is an important component of the methodology towards the sea thickness model but the author failed to mention what the trend was during the study period. It would be good to know how this compares with the information presented in the discussion. Is there a threshold of growth once a maximum temperature is reached? Is that why maximum growth takes place between November to January and not after.

Here we present the air temperature data used for ice thickness models. The lowest temperatures are observed in January. This can explain why the significant growth was occurred between November to January. The air temperature data will be added if needed for a better explanation. Based on the ice thickness models, it is reached to the maximum in late May or early June, and then rapidly decreased. However, the proposed SPO technique is not applicable to the late stage of rapid changes and melting (i.e., surface melt and water over ice prevent radar penetration deeper). This was explained in lines 194-195. A follow-up study on the radar backscattering responses on the ice thickness growth and decay is ongoing, but it will be not included in this manuscript.

[Figure]

*Figure S2. Air temperature during 2017-2018 and 2018-2019 cycles (upper, from the Pelly Island station) and sea ice thickness models (lower).*

Did the authors look at the growth and displacement of bottomfast ice during the study period; does it show comparable growth patterns between the years?

Bottomfast ice is very stable compared to floating ice. While floating ice shows very strong backscattering from the ice-water interface, bottomfast ice appears very dark as radar signals penetrating into grounded ice are mostly absorbed into ground (Figure 3c). These are explained in lines 113-117. Thus, we could not detect any vertical offsets from bottomfast ice.

Technical Corrections

The figures were well done and helped to explain the results. A few edits below:

Figure 1 A – elevation model on land appears to be discontinuous with the abrupt break in the elevation at the latitude line. I am also confused on the colour, do they represent a range? (ie. 0-10, 10-50, 50-100). I think the map would look more realistic if you tighten the range of elevations at the lower levels (ie. 0-2, 2-5, 5-10, 10-50, etc) Bathymetry estimates seem believable.

It will be replaced with the following figure.

[Figure]

*Figure S3.*

Figure 1: (caption): missing bracket after b, adjust other brackets in caption to equal out.

To be checked as per the journal format.

Figure 5b: Please explain the large offset arrow coming from the Yukon coast in the Shingle Point area.

It is considered to be a significant horizontal displacement of floating ice off from the coast caused by the strong winds from east and south during 20180125-20180206.

Figure 5d: The red circles are hard to see

The colors will be adjusted for better visualization

Figure 6: Perhaps the land value can be black which is instantly distinguishable and matches the circle in Figure 5.

The green color was chosen to match with the vertical motions of the land represented in green in Fig. 5.

Line 35: Does bathymetry play a role in the distribution of landfast ice – perhaps it should be mentioned here. The shallow water depth of the Mackenzie Estuary definitely plays a role in the distribution of bottomfast ice and subsequent water discharge through these regions defined by the bottomfast ice barrier.

Agreed. The recurrent pattern of landfast ice is controlled by the shallow bathymetry (Nghiem et al., 2014). An explanation for the bathymetry effect will be added here.

Line 129: Please explain what an orbital phase ramp is?

It is 'a residual phase induced by orbital differences or errors'. To be added.

Line 158: Please add reference

'Source from the Mackenzie River Middle channel (10MC008) available at http://wateroffice.ec.gc.ca' will be added.

Line 160: I think it would be good to insert some stats on prevenient wind direction from the Pelly station. Line 168: Yes agree, tidal fluctuations are nil but wind driven storm surge (even in the winter) have been known to cause over ice flooding. It would be good to know what the wind pattern was during the study period.

As explained above, Figure S1 will be added in the revised manuscript to explain the wind effects

Line 171: Chris Stevens (University of Calgary 2006-2011) conducted an MSc and PhD looking at the this area through GPR, perhaps his subsequent publication would provide some validation. May I also suggest the GSC online expedition database ED at Sea for further validation? https://ed.gdr.nrcan.gc.ca/index_e.php search cruise 2007301, Solomon describes a series of ice measurements with thickness and bottomfast ice

Thanks for your great suggestion. We used his paper and PhD thesis (Stevens et al. 2009; Stevens, 2011) on GPR ice thickness measurements and very low salinity measurements near the Mackenzie offshore in the reference list. The GPR measurements of the floating ice correspond to our estimations between 1.5 to 2 m though they were measured in 2005 and 2006. Unfortunately, we only found very brief descriptions on the 2007301 expedition from the GSC database.

References

Nghiem, S. V., Hall, D. K., Rigor, I. G., Li, P., and Neumann, G.: Effects of Mackenzie River discharge and bathymetry on sea ice in the Beaufort Sea. Geophysical Research Letters, 41(3), 873–879. https://doi.org/10.1002/2013GL058956, 2014.

Stevens, C. W.: Controls on Seasonal Ground Freezing and Permafrost in the Near-shore Zone of the Mackenzie Delta, NWT, Canada. PhD thesis, University of Calgary, https://doi.org/10.11575/PRISM/15339, 2011.

Stevens, C. W., Moorman, B. J., Solomon, S. M., and Hugenholtz, C. H.: Mapping subsurface conditions within the near-shore zone of an Arctic delta using ground penetrating radar. Cold regions science and technology, 56(1), 30-38, https://doi.org/10.1016/j.coldregions.2008.09.005, 2009.

---

## Author Comment (AC2) · 5 Aug 2020

**Responses to RC2**

**1. No explanation of why backscatter from ice bottom would remain coherent during growth**

The speckle offset tracking technique requires that the scattering surface remains coherent between image acquisitions. However, in the case of scattering from the bottom of a growing floating ice cover, the radar is seeing an entirely new surface at each acquisition and I therefore see no reason why the speckle would remain consistent over timespans of 12 days. The authors need to provide more explanation of how the scattering characteristics of the underside of the sea ice (if indeed that is where the signal is coming from) would remain constant as new ice forms below each previously imaged surface.

The offset tracking algorithm seeks the peak in the 2D cross-correlation between master and slave images. This peak does not have to be large (which corresponds to high coherence), but it has to be distinct (i.e. distinct global maximum). This can be achieved by using a very large window when computing the 2D cross-correlation. In our study, this is achieved by a search window of 256x64 pixels. The resolution of Sentinel-1 SAR data is about 2.3x13.9 m, so the search window is ~596x888 m or 0.5 km2. We claim that the predominant scattering over the area of 0.5 km2 comes from the ice/water interface. This is, however, not the only scattering surface in 0.5 km2 resolution cell. Other scattering surfaces are at various ice depths (depending on the salinity of individual 2.3x13.9 m pixels), but they do not add up to a synchronous response. The synchronous response from the ice/water interface produces a global maximum of 2D cross-correlation. The peak in the 2D cross-correlation function produced by a large search window is small but distinct. For example, Figure S1 shows the 2D cross-correlation features by applying a small window of 64x64 pixels and a large window of 256x256 pixels for the same central pixel. As shown, a small search window (left) produces a number of peaks. Some of them correspond to the reflections from the air/ice interface that capture motions at the surface. When a large window is used (right), only one distinct peak remains. See also next comment for additional details.

Figure S1. 2D cross-correlation plots calculated with a small window (left) and a large window (right)

**2. Inadequately supported assumption that ice bottom is source of SAR backscatter**

The authors assume that the SAR signal from the stabilized floating landfast ice is coming from the icewater interface. The basis of this assumption is the low-backscatter signature of presumed bottomfast ice nearby and the presence of low-salinity ice in this region reported by MacDonald et al (1995). However, none of ice sampled by MacDonald et al was completely fresh and most contained a significant seawater fraction. Moreover, close inspection of the SAR intensity imagery (Fig 3d,e) shows linear spatial patterns typically associated with surface roughness features. It therefore seems likely that some fraction of the microwave energy returned from the ice is coming from its upper surface and volume. This has a significant bearing on the interpretation of the SPO results, but is not discussed in the manuscript.

In our previous response, we explained why scattering at ice/water interface is captured by the offset tracking algorithm. This is not the only scattering mechanism present in SAR image, other scattering from the air/ice interface and at various ice depths are also present. But, this scattering does not add up to a coherent response over an area of  $0.5 \text{ km}^2$  (i.e., window size for computing 2D cross-correlation in this study). Figure S2 shows the true-color composite of the Beaufort Sea, just north of the Mackenzie River Delta as observed by the MODIS. The brown and tan river sediments discolored the water and hinted at the extent of the outflow. It can be seen that freshwater discharged by rivers in this region propagates far from the shore affecting sea surface temperature (Nghiem et al., 2014), which supports our claim of lower salinity. MacDonald et al. (1995) confirmed from GI-1 and PI-1 stations near the shallow shore that the ice core samples are mostly composed of freshwater (e.g., freshwater ice/total ice=1.71 m/1.72 m for GI-1) and the salinity is maintained close to 0 after mid-November (see its Figure 12). Stevens (2011) also confirmed that pore water salinity measurements (e.g., drilling sites BH01-BH05) are close to 0 up to ~ 5m depth below ice surface.